


# Reduced light absorption of black carbon (BC) and its influence on BC-boundary-layer interactions during "APEC Blue"

Meng Gao[1,2], Yang Yang[1], Hong Liao[1], Bin Zhu[3], Yuxuan Zhang[4], Zirui Liu[5], Xiao Lu[6], Chen Wang[7], Qiming Zhou[2], Yuesi Wang[5], Qiang Zhang[8], Gregory R. Carmichael[7], Jianlin Hu[1]

[1]Collaborative Innovation Center of Atmospheric Environment and Equipment Technology, Jiangsu Key Laboratory of Atmospheric Environment Monitoring and Pollution Control (AEMPC), Nanjing University of Information Science & Technology, Nanjing 210044, China.
[2]Department of Geography, State Key Laboratory of Environmental and Biological Analysis, Hong Kong Baptist University, Hong Kong SAR, China
[3]Key Laboratory for Aerosol-Cloud-Precipitation of China Meteorological Administration, Nanjing University of Information Science & Technology, Nanjing 210044, China
[4] School of Atmospheric Sciences, Nanjing University, Nanjing 210023, China
[5]State Key Laboratory of Atmospheric Boundary Layer Physics and Atmospheric Chemistry, Institute of Atmospheric Physics, Chinese Academy of Sciences, Beijing 100029, China
[6]Harvard John A. Paulson School of Engineering and Applied Sciences, Harvard University, Cambridge, MA, USA
[7]Department of Chemical and Biochemical Engineering, The University of Iowa, Iowa City, IA 52242, USA
[8]Ministry of Education Key Laboratory for Earth System Modeling, Department of Earth System Science, Tsinghua University, Beijing, 100084, China

*Correspondence to:* Meng Gao (mmgao2@hkbu.edu.hk) and Jianlin Hu (jianlinhu@nuist.edu.cn)

**Abstract.** Light absorption and radiative forcing of black carbon (BC) is influenced by both BC itself and its interactions with other aerosol chemical compositions. Although the changes in BC concentrations in response to emission reduction

measures have been well documented, the influence of emission reductions on the light absorption properties of BC and its influence on BC-boundary-layer interactions has been less explored. In this study, we used the online coupled WRF-Chem model to examine how emission control measures during APEC affect the mixing state/light absorption of BC, and the associated implications for BC-PBL interactions. We found that both the mass concentration of BC and the BC coating materials declined during the APEC week, which reduced the light absorption and light absorption enhancement ($E_{ab}$) of

BC. The reduced absorption aerosol optical depth (AAOD) during APEC were caused by both the declines in mass concentration of BC itself (52.0%), and the lensing effect of BC (48.0%). The reductions in coating materials (39.4%) dominated the influence of lensing effect, and the reduced light absorption capability ($E_{ab}$) contributed 3.2% to the total reductions in AAOD. Reduced light absorption of BC due to emission control during APEC enhanced planetary boundary layer height (PBLH) by 8.2 m. Different responses of $PM_{2.5}$ and $O_3$ were found to the changes in light absorption of BC.

Reduced light absorption of BC due to emission reductions decreased near surface $PM_{2.5}$ concentrations but enhanced near surface $O_3$ concentrations in the North China Plain. These results suggest that current measures to control $SO_2$, $NO_x$, etc. would be efficient to reduce the absorption enhancement of BC, and to inhibit the feedback of BC on boundary layer. Yet



enhanced ground $O_3$ might be a side effect of current emission control strategies. How to control emissions to offset this side effect of current emission control measures on $O_3$ should be an area of further focus.


## 1 Introduction

Black carbon (BC) in the atmosphere is produced both naturally and by human activities, attributable to the incomplete combustion of hydrocarbons (*Bond et al., 2013; Ramanathan and Carmichael, 2008*). In addition to contributing considerably to particulate matter and degraded air quality, it is the dominant absorber of visible solar radiation, playing a
unique and pivotal role in the Earth's climate system (*Bond et al., 2013; Menon et al., 2002; Ramanathan and Carmichael, 2008; Yang et al., 2019*). The absorption of BC occurs not only in the atmosphere, but when it is deposited over snow or ice, it triggers positive feedbacks and exert a positive radiative forcing (*Flanner et la., 2007; Grieshop et al., 2009*). The direct radiative forcing of atmospheric black carbon was estimated to be 0.4 W m$^{-2}$ (0.05-0.8 W m$^{-2}$) (*IPCC, 2014*), and BC has been targeted in emission control policies to mitigate both air pollution and global warming (*Grieshop et al., 2009*).

Before the 1950s, intense emissions of BC were concentrated in North America and Western Europe. In recent decades, South and East Asia have emerged to become major source regions (*Ramanathan and Carmichael, 2008*). BC emitted from China is responsible for a quarter of the total global emissions (*Bond et al., 2004*). Chemical transport model simulations suggest that the residential sector is the leading source for mass concentration of BC in China, followed by the industrial sector (*Li et al., 2016*). Mean BC direct radiative forcing in China is ~1.22 W m$^{-2}$, more than three times the global
mean forcing (*Li et al., 2016*), with two-thirds to three fourths of which contributed by local emissions of BC in China, and the rest by emissions in other countries (*Li et al., 2016; Yang et al., 2017*).

Specific policies to address BC emissions have not been implemented in China, yet multiple measures targeting PM$_{2.5}$ reduction have resulted in declines in BC (*Gao et al., 2018b; Yamineva and Liu, 2019*). A number of observational studies have revealed the declining trend of BC concentrations in China in recent years (*Ji et al., 2018, 2019a, 2019b; Qin et*
*al., 2019*). From 2013 to 2018, the annual mean BC concentrations in Beijing declined from 4.0 µg m$^{-3}$ to 2.6 µg m$^{-3}$ (*Ji et al., 2019b*). Associated changes in BC radiative forcing can be expected from declines in mass concentration of BC in China, while the radiative forcing of BC is influenced also by the changes in other aerosol components.

BC absorption is closely connected with the aging process, which is defined as the interaction between BC and other aerosol chemical compositions (*Jacobson, 2001*). After being emitted from combustion processes, BC particles can
coagulate and grow by condensation, during which both self-coagulation and hetero-coagulation happen (*Jacobson, 2001*). Although BC is mixed internally with other components, the system is impossible to be well-mixed due to the irregular shape of BC (*Jacobson, 2000*). A core-shell morphology is commonly established, with BC as the core and the coating materials (organics, sulfate, etc.) as the shell (*Jacobson, 2001; Zhang et al., 2018*). Numerous efforts have been made to explore the influence of aerosol components on internally mixed BC absorption (*Cappa et al., 2012; Chen et al., 2021; Bond*





*et al., 2006; Fuller et al., 1999; Jacobson, 2001; Liu et al., 2017; Peng et al., 2016*). It was proposed that the coating components (shell) could act as a lens to focus more photons onto the core to enhance the light absorption of BC (*Fuller et al., 1999*). *Bond et al. (2006)* estimated that this lensing effect would increase the light absorption of BC by 50-100%. *Jacobson (2001)* reported a global average BC absorption enhancement factor of 2, whereas other values, from negligible (*Cappa et al., 2012*) to as high as 2.4 (*Peng et al., 2016*) were also found previously. This lensing effect has been recognized

also as an important factor affecting radiative forcing of BC (*Jacobson, 2001*).

Over the past several years, the State Council of China has issued a comprehensive Air Pollution Prevention and Control Action Plan (APPCAP), covering major emission sectors (*Zhang et al., 2019a*). Long-term observations of aerosol chemical composition indicate that both concentrations of BC and other coating components have declined rapidly (*Gao et al., 2020b; Ji et al., 2019b; Zhou et al., 2019*). Although the changes in BC concentrations in response to emission reduction

measures have been documented (*Ji et al., 2019b; Gao et al., 2020b*), the influence of emission reductions on the aging processes and light absorption of BC has been less explored (*Zhang et al., 2019b*). *Zhang et al. (2018)* observed that the declines in absorption of BC was mainly dominated by decreases in BC mass concentration (86%), and the weakening of BC light absorption capability also played a role (14%). However, this finding was formulated based on surface observations, little is known about the changes at upper layers. Given the importance of BC absorption in the upper boundary layer as to

buildup of pollution (*Ding et al., 2016*), the impact of emission reductions on the light absorption of BC, and its implications for the development of boundary layer and pollution episodes need further investigations. On November 5-11, 2014, Beijing, China hosted the Asia-Pacific Economic Cooperation (APEC) meeting, during which Beijing and surrounding regions cooperated to implement short-term emission control measures to ensure good air quality. This event offers a great opportunity to study physical and chemical responses of atmospheric composition to emission reductions.

In this study, we address the following questions using the APEC event as a study case: (1) how did emission reductions affect the aging processes and light absorption of BC during APEC; (2) what were the relative contributions of reduced mass concentrations of BC, aging processes of BC, and reshaped mixing state of BC to the changes in light absorption of BC during APEC; and (3) how did these processes affect BC-PBL interactions and formation of air pollution? In Sect. 2, we describe the WRF-Chem model configurations and observational datasets used in this study. Results are

presented in Sect. 3, and conclusions/discussions are provided in Sect. 4.

## 2 Methods and data

### 2.1 WRF-Chem model configuration

WRF-Chem model (*Grell et al., 2005*) version 3.8.1 was adopted in this study to simulate emission, chemical

transformation and deposition of aerosols, as well as their interactions with radiation. We demonstrated in previous studies



(*Gao et al., 2016a, 2016b, 2020b, 2020c*) that the spatio-temporal variations of air pollutants over China could be reproduced well by WRF-Chem. WRF-Chem enables multiple options for gas phase chemistry and aerosol modules (*Grell et al., 2005*). We employed the Carbon Bond Mechanism version Z (CBMZ) gas phase chemistry (*Zaveri and Peters, 1999*) coupled with the Model for Simulating Aerosol Interactions and Chemistry (MOSAIC) (*Zaveri et al., 2008*) aerosol module

in this study. MOSAIC treats size resolved aerosol species, and we used 8 bins version in this study, corresponding to the particle diameter ranges of 0.039-0.078, 0.078-0.156, 0.156-0.312, 0.312-0.625, 0.625-1.25, 1.25-2.5, 2.5-5.0, 5.0-10.0 µm, respectively. Secondary organic aerosol (SOA) formation in MOSAIC was simulated with volatility basis set (VBS) (*Shrivastava et al., 2011*). We configured two nested domains with horizontal resolutions of 81km and 27km, and 31 vertical layers up to a pressure level of 50hPa. The configured domains cover most areas of East Asia and focus on the North China

region (same as Figure 1 in *Gao et al., 2017*). Other chosen options for key physical parameterizations follow *Gao et al. (2016b)*. Meteorological initial and boundary conditions were provided by the NCEP 1°×1° degree final reanalysis dataset (FNL), and chemical initial and boundary conditions were obtained from the MOZART global chemistry simulations (*Emmons et al., 2010*). To allow the effects of aerosol on meteorological conditions in the model, we did not apply observational nudging or reanalysis nudging.

Anthropogenic emissions of particles and gases in China in the model were taken from the Multi-resolution Emission Inventory for China (MEIC) for year 2014 developed by Tsinghua University (*Zheng et al., 2018*). Anthropogenic emissions for areas outside China were obtained from the MIX Asian emission inventory developed for MICS-Asia and HTAP, which combines five emission inventories for Asia (*Li et al., 2017*). Both MEIC and MIX datasets provide monthly emissions of air pollutants at 0.25°×0.25° grids, which were interpolated to WRF-Chem modeling domains in this study. We

adopted the MEGAN model version 2.04 to estimate biogenic emissions of gases and particles online (*Guenther et al., 2006*). The Global Fire Emissions Database version 4 (GFEDv4) (*Giglio et al., 2013*) were used as open fire emissions.

We simulated the period from October 16 to November 13, and discarded the first seven days as spin-up to avoid the influences of initial conditions. To explore the influences of coordinated emission control measures on BC absorption, we conducted multiple sets of simulations, as described in Table 1. For the NOCTL experiments, simulations were

conducted with no perturbations in emissions. For the CTL experiments, emissions of $SO_2$, $NO_x$, $PM_{10}$, $PM_{2.5}$, VOCs, and other species in Beijing were reduced by 39.2%, 49.6%, 66.6%, 61.6%, 33.6%, and 50%, respectively, over November 3-11 period. Emissions in Inner Mongolia, Shanxi, Hebei, Tianjin, and Shandong were reduced by 35%. These perturbation factors were taken from the BMEPB reports (*Gao et al., 2017*). The locations of these provinces are marked in Figure 1 in *Gao et al. (2017)*.

The influences of BC absorption under different assumptions, including external/core-shell mixing and with/without emission reductions ($\Delta_{BC-Ext-NOCTL}$, $\Delta_{BC-Ext-CTL}$, $\Delta_{BC-CS-NOCTL}$, $\Delta_{BC-CS-CTL}$, and $\Delta_{BC-CYSN-CS}$), can be derived with equations (1-5) below. The description of each simulation is documented in Table 1.

$$\Delta_{BC-Ext-NOCTL} = NOCTL_{Ext} - NOCTL_{Ext-nobc} \quad (1)$$

$$\Delta_{BC-Ext-CTL} = CTL_{Ext} - CTL_{Ext-nobc} \quad (2)$$






$$\Delta_{BC-CS-NOCTL}= NOCTL_{CS} - NOCTL_{CS-nobc} \ (3)$$

$$\Delta_{BC-CS-CTL}= CTL_{CS} - CTL_{CS-nobc} \ (4)$$

$$\Delta_{BC-CS-CYSN}= CYSN_{CS} - CYSN_{CS-nobc} \ (5)$$

The influences of emission reductions during APEC on changes in light absorption of BC and associated changes in meteorological/pollution conditions under external/core-shell mixing assumptions ($\Delta_{emission-Ext}$ and $\Delta_{emission-CS}$) can be

inferred with equations (6-7) below. We use equation (8) to derive the impact of changed BC aging processes by comparing the differences between core-shell simulation and external mixing simulation. The influences of reduced coating due to emission control measures during APEC are calculated with equation (9). We use equation (10) to derive the influences of changes in light absorption enhancement ($E_{ab}$) of BC.

$$\Delta_{emission-Ext}= \Delta_{BC-Ext-CTL} - \Delta_{BC-Ext-NOCTL} \ (6)$$


$$\Delta_{emission-CS}= \Delta_{BC-CS-CTL} - \Delta_{BC-CS-NOCTL} \ (7)$$

$$\Delta_{aging}= \Delta_{BC-CS-CTL} - \Delta_{BC-CYSN-CS} \ (8)$$

$$\Delta_{coating}= \Delta_{BC-CS-CYSN} - \Delta_{BC-CS-CTL} \ (9)$$

$$\Delta_{E_{ab}}= \frac{\frac{\Delta_{emission-CS}}{\Delta_{BC-CS-NOCTL}} - \frac{\Delta_{emission-Ext}}{\Delta_{BC-Ext-NOCTL}}}{\frac{\Delta_{emission-CS}}{\Delta_{BC-CS-NOCTL}}} \ (10)$$

## 2.2 Calculation of aerosol optical properties in WRF-Chem


WRF-Chem uses Mie theory to calculate layer aerosol optical depth (AOD), single scattering albedo (SSA), and asymmetry factor (g). First, the size parameter and spectral refractive index are used to calculate the Mie extinction efficiency $Q_e$. Then, the extinction coefficient $\sigma_e(\lambda)$ is provided by the integral of $Q_e$ with consideration of the geometric size of the particle ($\pi r^2$) and the particle number size distribution n(r) (equation (11)). $\sigma_e(\lambda)$ is a equation of wavelength $\lambda$.

Similarly, absorption coefficient $\sigma_a(\lambda)$ and scattering coefficient $\sigma_s(\lambda)$ can be obtained with Mie absorption efficiency $Q_a$ and Mie scattering efficiency $Q_s$. The value of SSA can be calculated with equation (12) using $\sigma_a(\lambda)$ and $\sigma_s(\lambda)$.

$$\sigma_e(\lambda) = \int_{r_{min}}^{r_{max}} Q_e \ \pi r^2 n(r) dr \ (11)$$

$$SSA(\lambda) = \frac{\sigma_s(\lambda)}{\sigma_s(\lambda)+\sigma_a(\lambda)} \ (12)$$

The calculated optical properties vary with the assumption of mixing state of aerosols. For external mixing, each

particle is assumed to be a single chemical species. There are several models proposed for internal mixing, and the commonly used ones include the volume averaging model and core-shell model. In the volume averaging model, all species are assumed to be well mixed, while the core-shell model assumes that BC is coated by a well-mixed shell of other species (*Jacobson, 2001*). The volume-weighted refractive index $m$ is obtained with the equation below:



$$m = \frac{\sum_i V_i m_i}{\sum_i V_i} \quad (13)$$

In equation (13), $V_i$ denotes the volume of species $i$ and $m_i$ represents the refractive index of species $i$. The official version of WRF-Chem does not calculate optical properties of aerosols with external mixing assumption. To assess the influence of aging process on the light absorption of BC, estimated light absorption of BC with external mixing assumption is required. We modified the optical calculation module in WRF-Chem so that it does not mix BC with other chemical species in the calculation of optical properties, which can be used as optical properties of BC with external mixing

assumption.

### 2.3 Observations

Both observations of meteorological variables and air pollutants were used to evaluate the performance of model over the APEC study period in *Gao et al. (2017)* and in this study. The meteorological measurements were retrieved from

the National Centers for Environmental Information website (https://gis.ncdc.noaa.gov/maps/ncei#app=cdo), which includes near surface temperature, relative humidity (RH), wind speed, and wind direction. The hourly surface concentrations of $PM_{2.5}$ and daily $PM_{2.5}$ chemical compositions were measured at the Institute of Atmospheric Physics (IAP), Chinese Academy of Sciences (CAS) site (*Liu et al., 2017; Yang et al., 2020*). We obtained also AAOD (absorption aerosol optical depth) from the AERONET network (*Dubovik and King, 2000; Holben et al., 1998*) to evaluate model performance. Data

from more than 500 sites across the world are provided online at the AERONET website (http://aeronet.gsfc.nasa.gov).

## 3 Results

### 3.1 Model Evaluation

Model evaluation was conducted with surface observations of meteorological variables, $PM_{2.5}$, $PM_{2.5}$ chemical

components, and AAOD. Data at two meteorological sites in urban Beijing were averaged, and were compared against the model values for the domain grid cell containing the monitoring site. Figure 1 indicates that the daily mean temperature and relative humidity (RH) are captured well by the model. Observed strong wind conditions are slightly underestimated, which is a common issue due to inaccurate land use inputs or other problems in the model (*Gao et al., 2018a*). Our previous investigation (*Gao et al., 2017*) suggested that temperature and RH were lower and northerly winds became more frequent

from before APEC to during APEC periods, contributing to pleasant air quality. Figure 2(a) displays the simulated and observed hourly $PM_{2.5}$ concentrations in urban Beijing. Before APEC, observed high $PM_{2.5}$ concentration is well captured by our model. During APEC, the NOCTL case overestimates $PM_{2.5}$ concentrations, while the CTL case exhibits better



agreement with observations. Implementing emission reductions in the model lowers the mean bias of the model from 30.8 to -4.0 µg m$^{-3}$. The performance of WRF-Chem in simulating wintertime PM$_{2.5}$ chemical compositions was explored extensively in our previous investigations (*Gao et al., 2016b, 2018a*). Similarly, measured high concentrations of inorganic aerosols (sulfate, nitrate and ammonium) are underestimated, which could be partly due to missing sulfate formation pathways (*Cheng et al. 2016*). We used the updated version with heterogeneous sulfate formation (*Gao et al., 2016a*) to reduce the underestimation of sulfate in this study. Simulated BC concentration shows high degree of consistency with observations, while OC is slightly underestimated due to large uncertainties in current status of SOA modeling (Figure 2f). In general, the temporal variations and magnitudes of air pollutants are well represented in our model. Figure 1(d) compares simulated AAOD with external mixing assumption and core-shell model against AERONET inferred AAOD during the APEC study period. AAOD simulated with external mixing assumption exhibits much lower values than observation. With the core-shell model, this underestimation is largely reduced. However, AAOD is still underestimated by the model, which might be caused by missing sources of absorbing particles in the model. Currently, the absorption of organics is not treated in the WRF-Chem model, which is likely to underestimate the light-absorbing capability of carbonaceous aerosols in the atmosphere (*Andreae and Gelencser, 2006*). Uncertainties in the aerosol size distribution in emissions may also contribute to this mismatch between the model and observations (*Matsui, 2006*).

**3.2 Reductions in the concentrations of BC/coating pollutants and changes in BC aging degree**

Previously, the reductions of air pollutants were estimated by comparing concentrations of air pollutants during the APEC period with those during other periods. Given the differences in meteorological conditions, such a comparison is not able to indicate the influence of emission control measures. As displayed in Figure 2(a), the concentrations during October 24-25 can be two times of those during October 26-27, although no emission reduction measures were implemented. Previously, we concluded that the meteorological conditions during the APEC week were generally favorable for good air quality compared to it during the week before the APEC week (*Gao et al., 2017*). Thus, we perturbated emissions in this study to examine how it would affect concentrations of air pollutants, including both BC and its coating pollutants. As displayed in Figure 3(a-b), mean concentrations of SO$_2$ and NO$_2$ in urban Beijing declined by 38.7% and 36.3%, respectively, in response to short-term emission control measures. Based on observations, *Zhang et al. (2018)* reported that SO$_2$ concentrations decreased by 35% (67%) and NO$_2$ concentrations decreased by 34% (45%) compared with that before (after) APEC. These declines in aerosol precursors would have modified secondary aerosol formation during the APEC week. Our model indicates that sulfate and nitrate declined by 40.0% and 28.2%, respectively. Given the slight underestimation of sulfate and nitrate, these values might have been moderately underestimated. Mass concentrations of BC declined by 34.6%, while the abundance of OC in the atmosphere exhibited a larger reduction (44.2%).





The changes in BC aging process is determined by both the decrease in BC and primary/secondary pollutants

condensed on BC. We used the ratio of the sum of pollutants (primary as well as secondary) to black carbon concentrations

($rBC$) to track the changes of BC aging degree:

$$rBC = \frac{[sulfate]+[nitrate]+[ammonium]+[organics]+[dust]+[sodimum]+[chloride]}{[BC]} \quad (14)$$

As shown in Figure 3(c), the impacts of emission reductions during APEC on $rBC$ behave differently at different

sizes. For ultrafine particles, emission reductions generally lower the aging degree of BC. This is consistent with the

observational evidence that smaller BC cores show larger reductions in aging degree as a result of emission control measures

during APEC (*Zhang et al., 2018*). As most secondary aerosols are in smaller sizes, the effect of emission reduction on BC

aging is more significant for smaller particles. *Zhang et al. (2018)* reported only the changes in sizes below 0.2 µm, our

modeling results suggest, however, that the aging degree of BC might be enhanced under emission reductions for relatively

larger particles (Figure 3c). The impact of emission reductions on $rBC$ behaves differently near the surface and at higher

layers (Figure 3d). The aging degree is lowered in the CTL case near the surface, mainly due to reductions in coating

materials. However, at layers higher than 200 meters, the aging degree of BC increases with emission reductions. In-situ near

surface measurements indicate also that $rBC$ was reduced during APEC, and the reduction was most likely caused by lower

photochemical production (*Zhang et al., 2018*).

### 3.3 Changes in AAOD and the light-absorption enhancement ($E_{ab}$) of BC during APEC

$rBC$ values describe the aging degree of BC, while the exploration of how emission reductions affect light absorption of BC

requires a sophisticated calculation of optical properties of BC. Mie theory is commonly used to calculate the light

absorption enhancement of BC ($E_{ab}$) from lensing effect with a core-shell model. *Zhang et al. (2018)* estimated $E_{ab}$ by

dividing the light-absorption cross section of the whole BC-containing particle by that of BC core at a certain wavelength.

Here we follow the method in *Curci et al. (2019)*, and calculate $E_{ab}$ as the ratio of BC AAOD estimated assuming core-shell

internal mixing to that calculated with external mixing assumption:

$$E_{ab} = \frac{BC\_AAOD(550nm,core-shell\ mixing)}{BC\_AAOD(550nm,external\ mixing)} \quad (AAOD\ can\ be\ either\ layer\ or\ column) \quad (15)$$

Inferred vertical profiles of layer $E_{ab}$ values in the CTL and NOCTL scenarios are displayed in Figure 3e. At the layers

below 5km, mean $E_{ab}$ values are 1.96 and 1.95 for CTL and NOCTL scenarios, respectively. Below 500 m in the

troposphere, emission reductions during APEC lower $E_{ab}$ from 2.11 to 2.06. Previous study by *Jacobson (2001)* suggests a

global average BC absorption enhancement factor of 2, which is consistent with current study. However, a wide range of

enhancement factors have been reported, from negligible (*Cappa et al., 2012*) to as high as 2.4 (*Peng et al., 2016*). *Liu et al.

(2017)* pointed that the enhancement factors depend on the particles' mass ratio of non-black carbon to black carbon. Our

model results indicate also that the reductions in light absorption enhancement of BC are concentrated at lower layers, while





enhancement could happen at higher layers (Figure 3e). This is consistent with the vertical profile of $rBC$ where it decreases

due to emission reductions near the surface while increases at higher layers (Figure 3d).

Figure 4a presents the daytime mean (defined as the mean BC AAOD over 10:00-17:00 time period) BC AAOD in Beijing

inferred from simulations with different mixing assumptions and emission perturbations. In the NOCTL scenarios, BC

AAOD simulated with core-shell model exhibits higher values than those with external mixing assumption (0.0220 for

external and 0.0427 for core-shell). Due to reductions in emissions, these values decline to 0.0145 and 0.0283, respectively.

Due to emission reductions (differences between CTL and NOCTL scenarios), mean daytime BC AAOD decrease by 0.0075

during the APEC week, as a result of declines in mass concentration of BC (52.0%, Table 2). However, the lensing effect of

BC induces a further decline of 0.0069 (48.0%, Table 2). The influence of lensing effect is dominated by the reductions in

coating materials (39.4%, equation (9), Table 2). The BC absorption enhancement ($E_{ab}$) factor decreased by 0.003 due to

reductions in emissions (Figure 4b). We further quantified that the reduced light absorption capability ($E_{ab}$) resulting from

emission reductions during APEC contributed 3.2% to the total reductions in AAOD (equation (10), Table 2).

### 3.4 Influences on boundary layer process and air pollution

The vertical distribution of BC absorption plays an important role in modulating the temperature gradient and changing

boundary layer meteorology (*Ding et al., 2016*). We conducted a series of numerical experiments to understand the

influences of reshaped BC absorption due to emission reductions during APEC on boundary layer process and the formation

of air pollution. Figure 4c illustrates the vertical profiles of BC absorption induced changes in equivalent potential

temperature (EPT), which is commonly used to indicate the stability of air in the atmosphere (*Obremski et al., 1989*). When

EPT decreases with height, the atmosphere is unstable and vertical motion/convection is likely to occur. In all experiments,

BC absorption induces a positive impact on EPT in the air above ground acting to enhance the stability of the atmosphere

(Figure 4c). The maximum enhancement occurs at layers close to 1-2km (Figure 4c). At ~2.6km, the maximum ratio of

changes with core-shell model to those with external mixing reach above 2.5, indicating the important effects of mixing state

of BC in the upper boundary layer (Figure 4c).

In urban Beijing, BC absorption induced mean changes of daytime planetary boundary layer height (PBLH) during the

APEC week are -11.6 and -24.0 m for external mixing and core-shell model, respectively (Figure 4d). Under a relatively

clean condition (CTL scenarios), these values change to -8.8 and -15.6 m for external mixing and core-shell model for

NOCTL emissions (Figure 4d). Due to emission reductions, the impacts of BC absorption on PBL inhibition decrease by 8.2

m (reduced emissions enhance PBLH by 8.2 m). The influences of reduced mass concentration of BC itself account for 35%

of the total changes, while the lensing effect of BC explain the rest (65%, Table 2). The decreased coating due to emission

reductions dominate the lensing effect of BC (47.4%, Table 2).





The corresponding changes in daytime mean near surface concentrations of $O_3$ and $PM_{2.5}$ in Beijing are displayed in Figure 4e and Figure 4f, respectively. The inhibited development of PBL due to BC absorption results in higher abundance of $PM_{2.5}$ within the PBL (*Ding et al., 2016; Gao et al., 2016b*). Previously, we quantified that the co-benefits of reduced aerosol feedbacks could explain ~11% of the total decreases in $PM_{2.5}$ in Beijing during APEC. Here we focus on light absorption of

BC, and find that the lensing effect of BC decreases $PM_{2.5}$ concentration by 0.8 µg m$^{-3}$ on average (Figure 4f). On average, declines in BC mass concentration itself account for 64.3% of the total impact of reduced light absorption of BC on $PM_{2.5}$, while 35.7% is attributed to the lensing effect of BC. However, inhibited PBL development does not necessarily lead to enhanced levels of near surface $O_3$, as the formation of $O_3$ is also affected by changes in aerosols and photolysis reactions above the ground. As displayed in Figure 4e, near surface $O_3$ concentrations in urban Beijing decrease in response to BC

absorption.

The spatial distribution of $\Delta_{emission-Ext}$ and $\Delta_{emission-CS}$ in Figure 5 reveal that external mixing and core-shell models estimate similar patterns of changes in AAOD, PBLH, near surface $O_3$ and near surface $PM_{2.5}$. However, the responses of these variables are larger in the core-shell model due to lensing effects of coating materials. Reduced emissions of BC and its coating materials during APEC led to declined AAOD, less stabilized PBLH, decreased near surface $PM_{2.5}$ concentrations

and enhanced near surface $O_3$ concentrations in the North China Plain (Figure 5).

Figure 6a,6b illustrates the cross sections in the northeast direction of changes in BC absorption coefficient due to emission reductions, as Beijing and polluted cities in South Hebei are covered. Pronounced declines are concentrated below 2km, and the core-shell model estimates stronger reductions due to lensing effects (Figure 6c). Emission control measures also reshaped the light absorption enhancement factor of BC, as indicated in Figure 6f. Within the lower boundary layer, $E_{ab}$

values were reduced with emission reductions during APEC. Light absorption of BC stabilizes boundary layer to accumulate $PM_{2.5}$, yet this effect is inhibited at lower emission levels during APEC. These relationships are reflected in Figure 6g and 6h with negative changes in $PM_{2.5}$ near the ground.

The responses of $O_3$ to reduced light absorption of BC during APEC are in the opposite direction (*Gao et al., 2018c*), compared to those for $PM_{2.5}$. Strong absorption of BC tends to enhance photolysis above the aerosol layer, but to reduce

photolysis near the ground. Figure 7d, 7g illustrate the changes in $O_31D$ and $NO_2$ photolysis rates with emission reductions inferred from an external mixing assumption. With emission control implemented, photolysis rates near the ground are enhanced due to lower light absorption of BC, while the photolysis rates above the aerosol layer are reduced. Similar patterns but with larger values are found using the core-shell model (Figure 7e, 7h). The responses of $O_3$ are generally in line with the responses of $O_31D$ and $NO_2$ photolysis rates (Figure 7a, 7b).




## 4 Summary and Discussions

In this study, we used the online coupled WRF-Chem model to understand how emission control measures during the APEC event would affect the mixing state/light absorption of BC, and the implications for BC-PBL interactions. Multiple observations, including surface observations of meteorological variables, $PM_{2.5}$, $PM_{2.5}$ chemical composition, and
AAOD were used to evaluate model performance. A series of numerical experiments were conducted to address three questions: (1) how did emission reductions affect the aging processes and light absorption of BC during APEC; (2) what were the relative contributions of reduced mass concentrations of BC, aging processes of BC, and reshaped mixing state of BC to the changes in light absorption of BC during APEC; and (3) how did these processes affect BC-PBL interactions and formation of air pollution?

We found that both the mass concentration of BC and the BC coating materials declined during the APEC week, which reduced the light absorption and light absorption enhancement ($E_{ab}$) of BC. Below 500 m in the troposphere, emission reductions during APEC lowered the absorption enhancement factor $E_{ab}$ from 2.11 to 2.06. The column absorption enhancement was reduced also. The reduced AAOD during APEC is caused by both the declines in mass concentration of BC itself (52.0%) and the lensing effect of BC (48.0%). The reductions in coating materials (39.4%) dominated the influence
of lensing effect, and the reduced light absorption capability ($E_{ab}$) contributed 3.2% to the total reductions in AAOD. Our estimate of the contribution of reduced light absorption capability ($E_{ab}$) exhibit lower values than *Zhang et al. (2016)*, which could be caused by the uncertainties in the assumption of the mixing state of BC in the core-shell model.

The diminished light absorption of BC during APEC promotes the development of PBL, as indicated in the changes in vertical profiles of EPT. Different responses of $PM_{2.5}$ and $O_3$ were found to the changes in light absorption of BC. The
responses of $PM_{2.5}$ follow the enhanced PBLH to decrease, while $O_3$ concentrations increase near the ground. The enhanced levels of $O_3$ were mainly caused by the influences of BC absorption on photolysis rates. As displayed in the conceptual scheme plot in Figure 8, reduced emissions of BC and its coating materials during APEC led to declined AAOD, less stabilized PBLH, decreased near surface $PM_{2.5}$ concentrations and enhanced near surface $O_3$ concentrations in the North China Plain.

This study with perturbations of emissions during APEC offer important implications on the potential effects of China's Clean Air Act. As discussed in our previous investigation (*Gao et al., 2017*), emission control measures have the co-benefits of reducing aerosol feedbacks to accelerate the cleaning of air, which accounts for ~11% of the decreased $PM_{2.5}$ concentrations during APEC. In this study, we further clarified that the ongoing measures to control $SO_2$, $NO_x$, etc. would be efficient to reduce the absorption capability of BC to inhibit the feedback of BC on the boundary layer. Our results also show
that near ground $O_3$ responds differently from the changes in $PM_{2.5}$, which might be a side effect of current emission control strategies. *Ma et al. (2021)* reported that aerosol radiative effect could explain 23% of the total change in surface summertime $O_3$ in China. How to control emissions to offset this side effect of current emission control measures on $O_3$



should be an area of further focus. In addition to the influences on air quality and weather, a sudden reduction in aerosol emissions may potentially affect climate (*Ren et al., 2020; Yang et al., 2020*), which warrants further investigation.

Although careful validation was conducted in this study, uncertainties still remain in the current study. We concluded that the core-shell model captures the variation of AAOD better than external mixing assumption. However, the core-shell model is an ideal scenario that assumes all non-BC materials are internally mixed and coated on BC. *Zhang et al. (2016)* observed that BC particles are heavily coated and are in a near-spherical shape in the North China Plain. The usage of core-shell model seems to be reasonable in this study, whereas the assumption that all non-BC materials are coated on BC

might not be true in real atmosphere. The observed ratio of coatings to $PM_1$ was ~25-70% in summer in Beijing (*Xu et al., 2019*), and the observed ratio of coatings to $PM_{2.5}$ was ~10-40% in winter in Beijing (*Wang et al., 2019*). Thus, the assumption of BC coating in the model might have overestimated $rBC$ in this study, leading to uncertainties in the results. In the near future, we would examine how different assumptions of BC coating would affect the light absorption properties of BC. Additionally, the simulated feedbacks of BC absorption on boundary layer processes are not well constrained. We used

multiple coupled models to examine how these processes are represented, and we calculated ensemble mean to obtain the best current understanding (*Gao et al., 2018a, 2020a*). In the future, further efforts are needed also to constrain the uncertainties of these processes in the model.

Data availability. The measurements and model simulations data can be accessed through contacting the corresponding authors.

Author contributions. MG and JH designed the study. MG performed model simulations and analyzed the data with help from YY, HL, BZ, YZ, XL, CW, QZ and GRC. QZ provided the emission inventory. YW and ZL provided

measurements. MG and JH wrote the paper with inputs from all the other authors.

Financial support. This work was supported by the Open fund by Jiangsu Key Laboratory of Atmospheric Environment Monitoring and Pollution Control (KHK1902), the National Key Research and Development Program of China (grant no. 2016YFA0602003), the National Natural Science Foundation of China (no. 42005084 and no. 92044302), the

Ministry of Science and Technology of the People's Republic of China (Grant no. 2017YFC0210000), the Natural Science Foundation of Guangdong Province (no. 2019A1515011633), and special fund of the State Key Joint Laboratory of Environment Simulation and Pollution Control (grant no. 19K03ESPCT).



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



**Table 1: Descriptions of model simulations**

| Experiments | Descriptions |
| --- | --- |
| $NOCTL_{Ext}$ | No perturbations in emissions; assuming external mixing of BC. |
| $NOCTL_{Ext-nobc}$ | No perturbations in emissions; assuming external mixing of BC; assuming no absorption of BC. |
| $NOCTL_{CS}$ | No perturbations in emissions; calculating optical properties using core-shell assumption. |
| $NOCTL_{CS-nobc}$ | No perturbations in emissions; calculation of optical properties using core-shell assumption; assuming no absorption of BC. |
| $CTL_{Ext}$ | emissions are reduced during APEC; assuming external mixing of BC. |
| $CTL_{Ext-nobc}$ | emissions are reduced during APEC; assuming external mixing of BC; assuming no absorption of BC. |
| $CTL_{CS}$ | emissions are reduced during APEC; calculation of optical properties using core-shell assumption. |
| $CTL_{CS-nobc}$ | emissions are reduced during APEC; calculation of optical properties using core-shell assumption; assuming no absorption of BC. |
| $CYSN_{CS}$ | emissions of BC are reduced while emissions of other species are not during APEC; calculation of optical properties using core-shell assumption. |
| $CYSN_{CS-nobc}$ | emissions of BC are reduced while emissions of other species are not during APEC; calculation of optical properties using core-shell assumption; assuming no absorption of BC. |






**Table 2: The division of the impact of BC absorption into the impact of BC mass itself and BC mixing state**

| Influenced variables | BC mass itself | BC lensing effect | Reduced coating | Reduced $E_{ab}$ |
|---|---|---|---|---|
| AAOD | 52.0% | 48.0% | 39.4% | 3.2% |
| PBLH | 34.9% | 65.1% | 47.4% | - |
| PM$_{2.5}$ | 64.3% | 35.7% | - | - |
| O$_3$ | 49.1% | 50.9% | - | - |

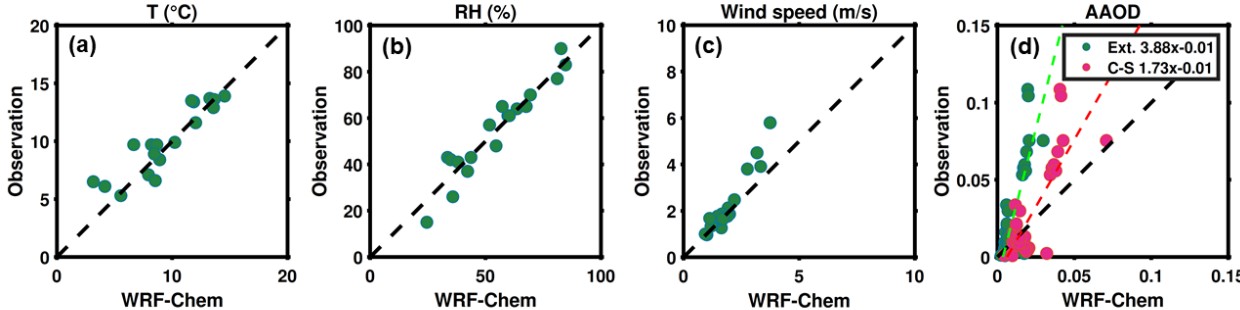

**Figure 1: Scatter plots of modeled and observed near surface meteorological variables (a: T, b: RH, c: wind speed); modelled AAOD with core-shell model/external mixing assumption, and the comparisons against observations.**



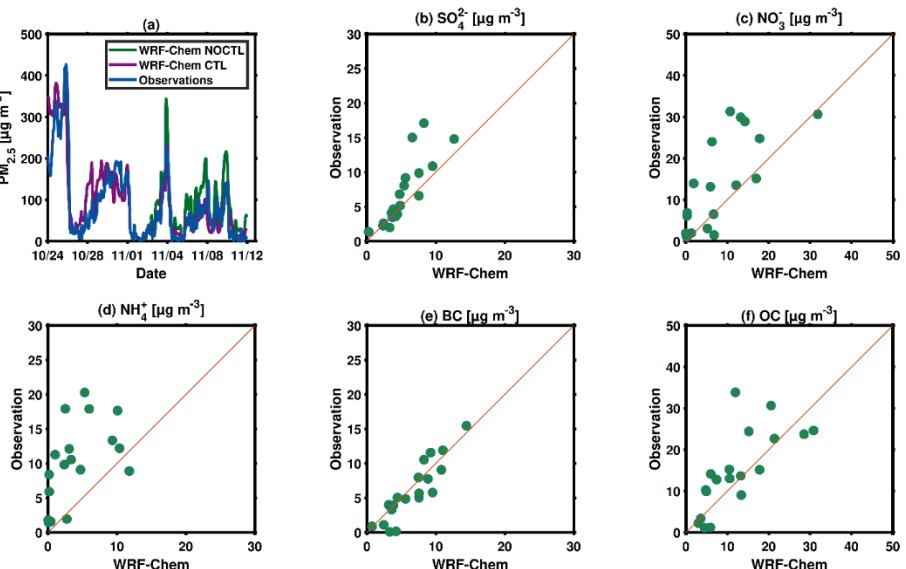

**Figure 2: Modeled and observed time series of PM₂.₅ concentrations in urban Beijing (a); Scatter plots of modeled and observed near surface daily mean concentrations of sulfate, nitrate, ammonium, BC (black carbon) and OC (organics) in Beijing (b-f).**

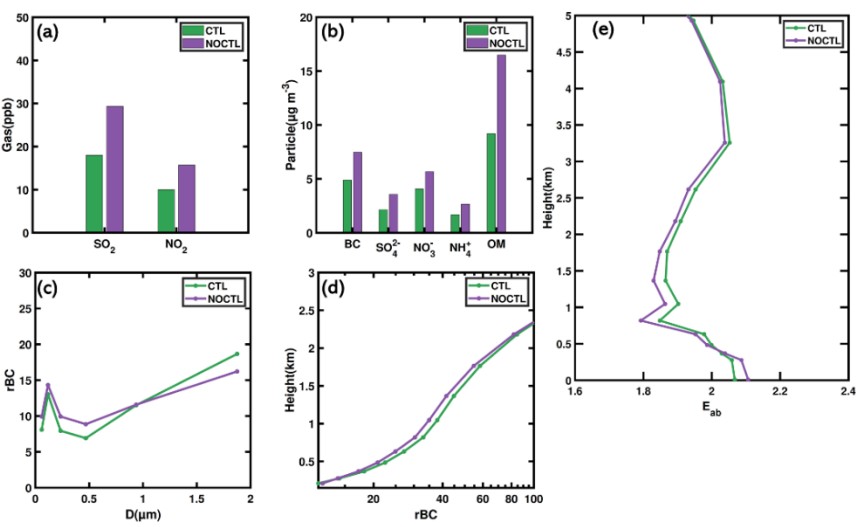

**Figure 3: Mass concentrations of gaseous and condensed pollutants in the CTL and NOCTL cases (a-b); the distribution of $rBC$ with sizes in the CTL and NOCTL cases (c); the distribution of $rBC$ with height in the CTL and NOCTL cases (d) and the distribution of $E_{ab}$ values with height in the CTL and NOCTL cases (e).**





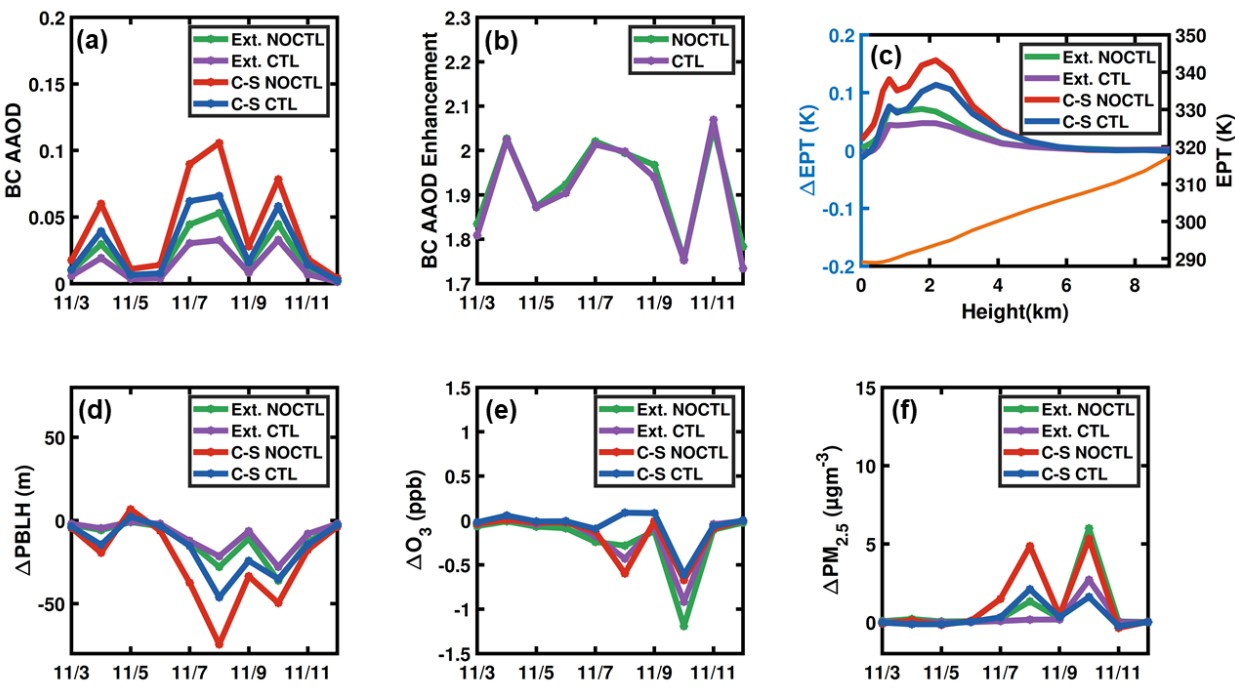

**Figure 4: Daytime (10:00-17:00 local time) mean BC AAOD in Beijing inferred from different simulations (a) and the BC AAOD enhancement in the NOCTL and CTL scenarios (b); BC absorption induced changes in EPT (orange line indicates the vertical profile of EPT) (c), PBLH (d), O₃ (e), and PM₂.₅ (f).**



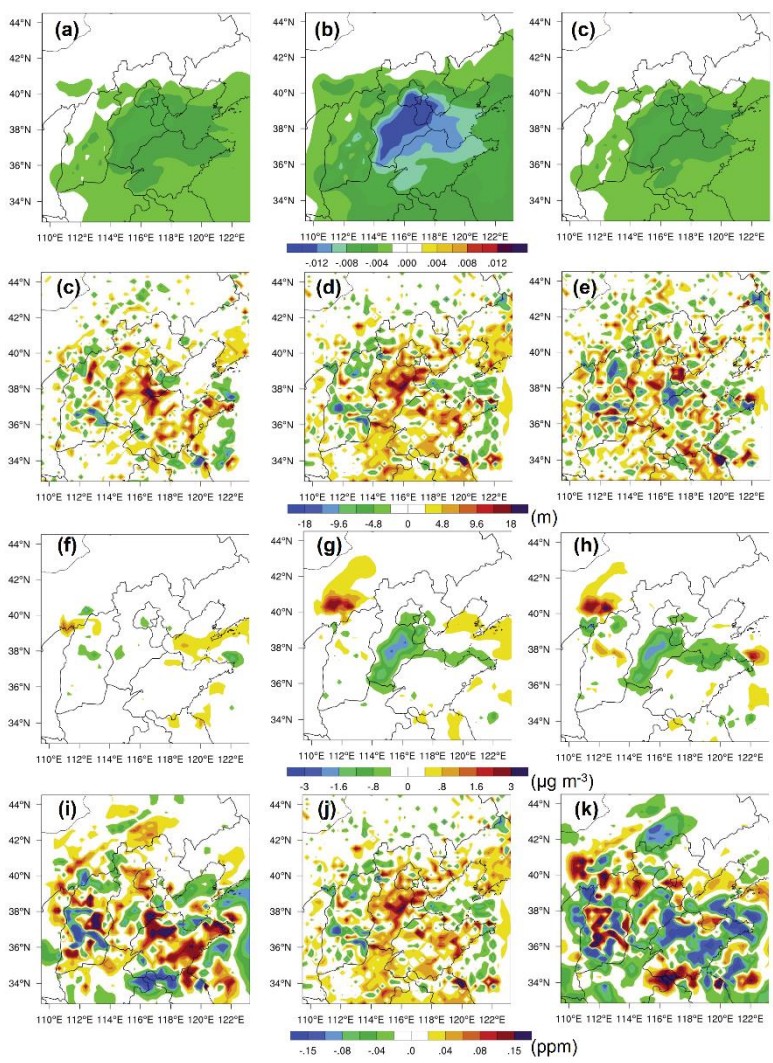

**Figure 5: Spatial distribution of daytime (10:00-17:00 local time) mean BC AAOD (first row), and mean BC absorption induced**
**changes after emission reductions (CTL minus NOCTL) in PBLH (second row), PM$_{2.5}$ (third row), and O$_3$ (fourth row); first,**
**second and third columns represent $\Delta_{emission-Ext}$, $\Delta_{emission-CS}$, and $\Delta_{aging}$.**


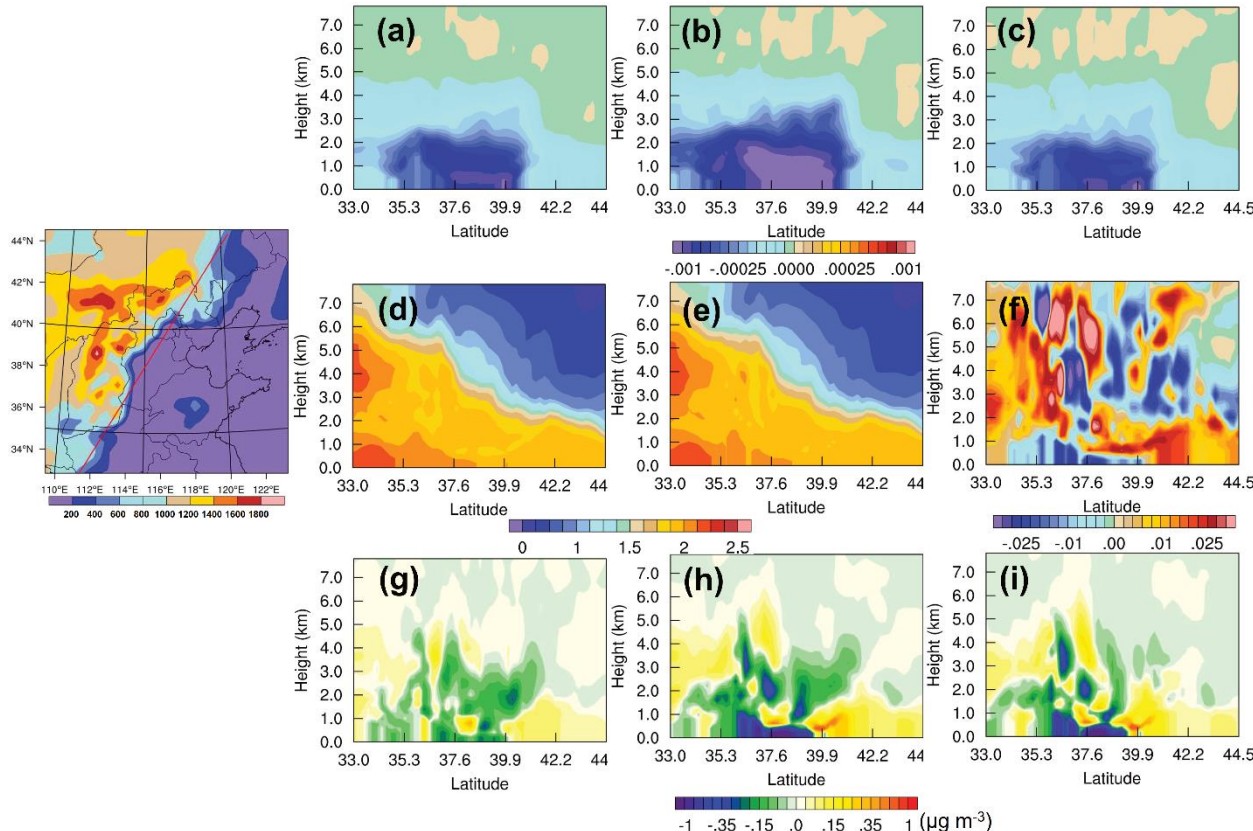

**Figure 6: Cross sections of daytime (10:00-17:00 local time) mean changes in BC absorption coefficient (first row, CTL minus NOCTL), $E_{ab}$ (second row), BC absorption induced changes after emission reductions (CTL minus NOCTL) in PM$_{2.5}$ (third row); first, second and third columns represent $\Delta_{emission-Ext}$, $\Delta_{emission-CS}$, and $\Delta_{aging}$.**

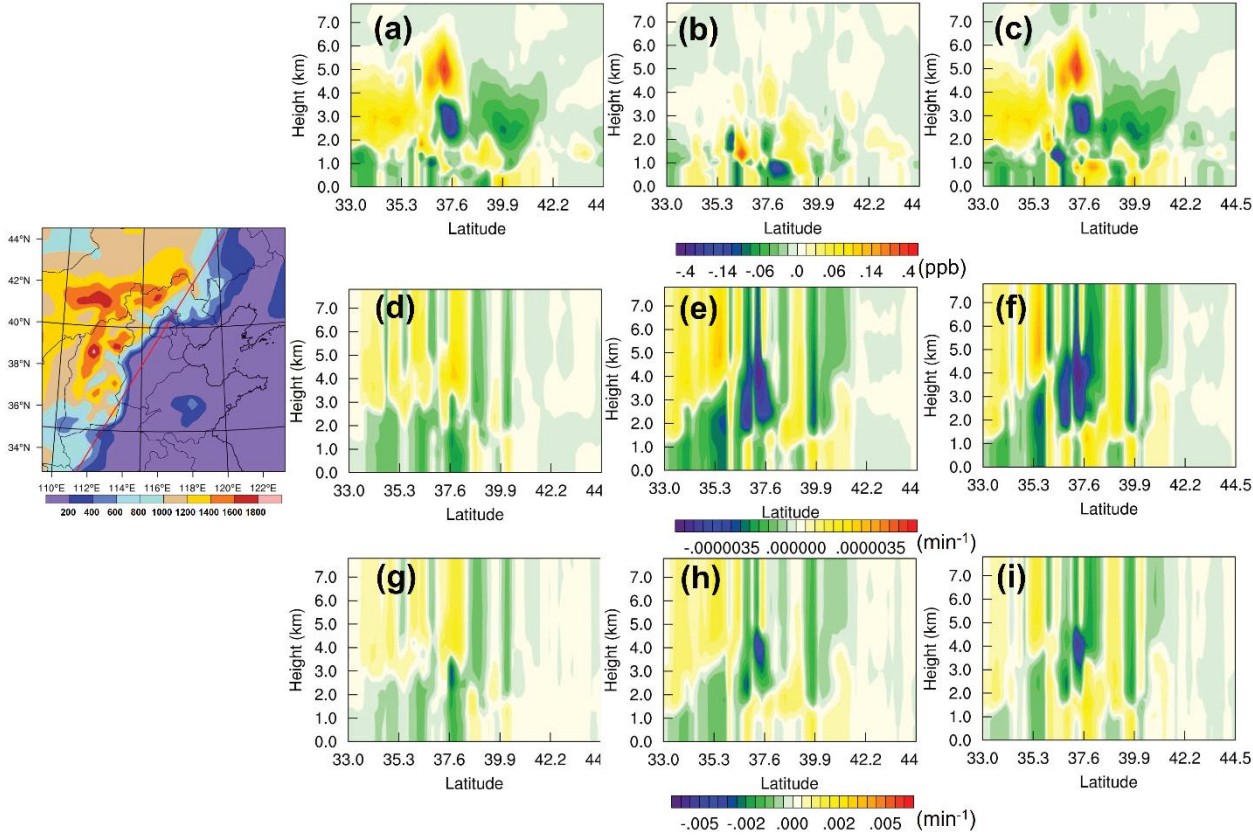

**Figure 7: Cross sections of daytime (10:00-17:00 local time) mean changes in BC absorption induced changes after emission reductions (CTL minus NOCTL) in O$_3$ (first row), O$_3$1D photolysis rate (second row), and NO$_2$ photolysis rate (third row); first, second and third columns represent $\Delta_{emission-Ext}$, $\Delta_{emission-CS}$, and $\Delta_{aging}$.**



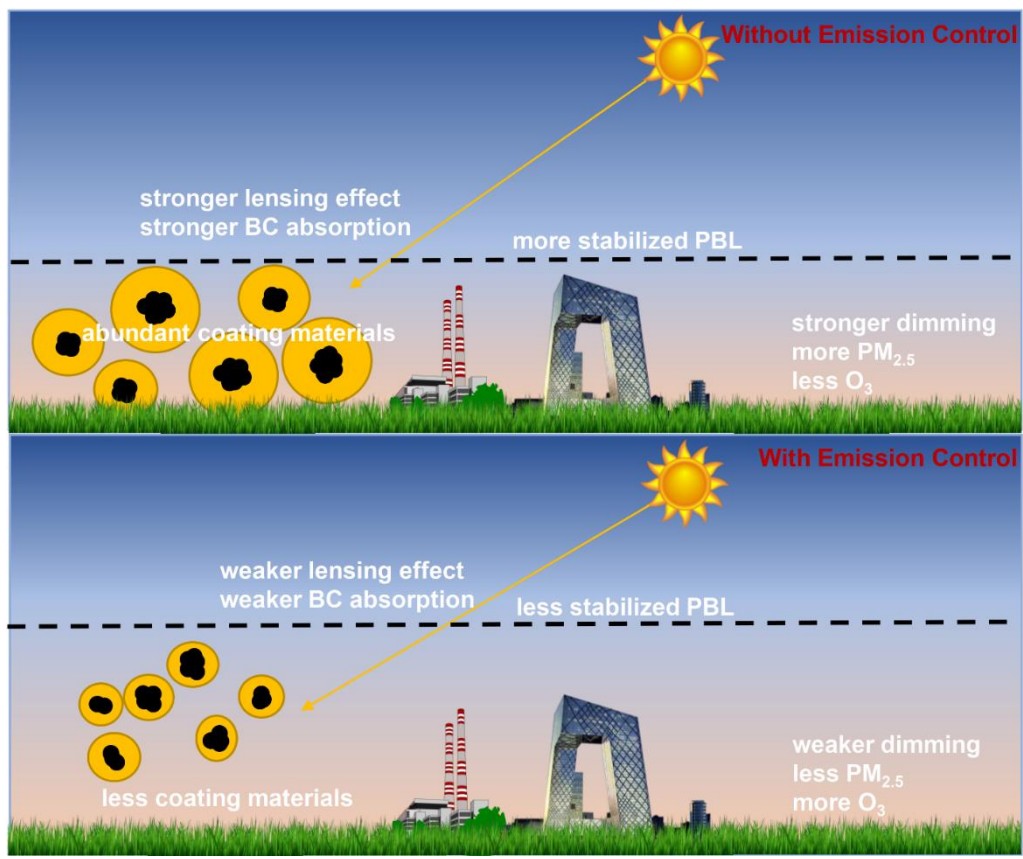

**Figure 8: Conceptual scheme of the effects of emission control during APEC on light absorption capability of BC, PBL and air quality.**


\