# Peer review of "Reduced light absorption of black carbon (BC) and its influence on BC-boundary-layer interactions during “APEC Blue”"

_Atmospheric Chemistry and Physics, 2021_

## Author Comment (AC1)

**#RC 1**

Black carbon (BC) is one of the major air pollutions severely threatening public health despite of its relatively low contribution to total PM mass, not only due to its toxicity but also its nature of light absorption. Because of this, BC is able to alter boundary layer stability and structure, which further influence the ventilation of pollutants. Previous studies have demonstrated the importance of this BC light absorption effect in surface layer pollution in many China megacities. This study use APEC Blue period as a natural lab to further quantify this impact using a fully coupled model. The scope fit well with ACP. The manuscript is well written, the results are scientific interesting and politically meaningful supported by sound methodology. I feel this work is suitable for publication after addressing a few minor concerns.

Reply:

➢ We thank the reviewer for careful reading and valuable suggestions, which are important to improve the quality of our manuscript.

**Minor concerns:**

1) Calculation of aerosol optical properties has been well described for the external mixture and internal homogeneous mixture (volume-weighted average). I feel the calculation details of core-shell mixture should also be elaborated, given core-shell is the main mixture style discussed in this paper. Please also provide the information of how complex reflective index is defined for each component, especially for organics.

Reply:

➢ Thanks for this suggestion. We added descriptions of core-shell calculation in the revised manuscript.
➢ "For core-shell internal mixing, similar averaging processes are applied to the core and shell separately. The scattering efficiency, absorption efficiency and asymmetry parameter are then obtained using the core-shell Mie theory documented in Toon and Ackerman (1981). Core-shell Mie calculation requests core radius, shell radius, refractive index of core and refractive index to shell as inputs (Toon and Ackerman, 1981)."

Toon, O. B. and Ackerman, T.P.: Algorithms for the calculation of scattering by stratified spheres, Appl. Opt., 20, 3657-3660, 1981.

➢ For reflective index for each component, we added Table S1 to list it, including OC.

Table S1. Complex refractive index for major aerosol components in WRF-Chem

| Components | real | imaginary |
|---|---|---|
| BC | 1.85 | 0.71 |
| Dust | 1.55 | 0.006 |
| Organics | 1.45 | 0 |
| $NH_4Cl$ | 1.50 | 0 |
| $NH_4NO_3$ | 1.50 | 0 |
| $NH_4HSO_4$ | 1.47 | 0 |
| $NA_2SO_4$ | 1.50 | 0 |

2) There are many ways/definitions of boundary layer top. How boundary layer top is defined and hence its height is estimated?

Reply:

➢ Different PBL schemes in WRF use different ways to determine the PBL top. For example, the MYJ scheme determines the PBL height using the TKE profile. It defines the top of the PBL to be the height where the TKE decreases to a prescribed low value (Janjic 2001). We used YSU scheme, and it defines the top of the PBL as the height where the bulk Richardson number calculated above the level of neutral buoyancy first exceeds a critical Richardson number (Hong et al., 2010).

Hong, S.Y., 2010. A new stable boundary-layer mixing scheme and its impact on the simulated East Asian summer monsoon. Quarterly Journal of the Royal Meteorological Society, 136(651), pp.1481-1496.

➢ We have added this information in the revised manuscript.

3) line 283. Do you mean reduce PBLH by 8.2m on average?

Reply:

➢ Yes, 8.2 on average.
➢ In the revised manuscript, we added the following description: "on average during the APEC week".

4) There is some nice discussion about the PBL-PM2.5-O3 interactions. I think some chemical reasons also influence ozone. Such as, reduce of PM2.5 could enhance the surface layer photolysis therefore increase ozone especially for heave polluted area/periods, and the co-reduction of NOx and the regime of NOx (Chen et al., 2021).

**References:**

Chen, Y., Beig, G., Archer-Nicholls, S., Drysdale, W., Acton, J., Lowe, D., Nelson, B. S., Lee, J. D., Ran, L., Wang, Y., Wu, Z., Sahu, S. K., Sokhi, R. S., Singh, V., Gadi, R., Hewitt, C. N., Nemitz, E., Archibald, A., McFiggins, G., and Wild, O.: Avoiding high ozone pollution in Delhi, India, Faraday Discussions, 10.1039/D0FD00079E, 2021.

Reply:

➢ Thanks for this suggestion. We discussed the influence of photolysis in the manuscript, as listed below.
➢ "However, inhibited PBL development does not necessarily lead to enhanced levels of near surface $O_3$, as the formation of $O_3$ is also affected by changes in aerosols and photolysis reactions above the ground. As displayed in Figure 4e, near surface $O_3$ concentrations in urban Beijing decrease in response to BC absorption."

- ➢ "The responses of $O_3$ to reduced light absorption of BC during APEC are in the opposite direction (Gao et al., 2018c), compared to those for $PM_{2.5}$. Strong absorption of BC tends to enhance photolysis above the aerosol layer, but to reduce photolysis near the ground. Figure 7d, 7g illustrate the changes in O31D and NO2 photolysis rates with emission reductions inferred from an external mixing assumption. With emission control implemented, photolysis rates near the ground are enhanced due to lower light absorption of BC, while the photolysis rates above the aerosol layer are reduced. Similar patterns but with larger values are found using the core-shell model (Figure 7e, 7h). The responses of O3 are generally in line with the responses of O31D and NO2 photolysis rates (Figure 7a, 7b)."
- ➢ Yes, we agree that changes in sources would also affect ozone. We interpret the changes with respect to optical properties, and emission levels were kept at the same level, as stated in equations (1-5). Thus, sources and chemical reasons would not affect our results.
- ➢ We added the discussions on chemical reasons in the manuscript: "$O_3$ is also affected by changes in aerosols and photolysis reactions above the ground (Chen et al., 2021)."

$$\Delta_{BC-Ext-NOCTL} = NOCTL_{Ext} - NOCTL_{Ext-nobc} \; (1)$$
$$\Delta_{BC-Ext-CTL} = CTL_{Ext} - CTL_{Ext-nobc} \; (2)$$
$$\Delta_{BC-CS-NOCTL} = NOCTL_{CS} - NOCTL_{CS-nobc} \; (3)$$

$$\Delta_{BC-CS-CTL} = CTL_{CS} - CTL_{CS-nobc} \; (4)$$

$$\Delta_{BC-CS-CYSN} = CYSN_{CS} - CYSN_{CS-nobc} \; (5)$$

5) Just curious that in Fig. 5h, why reduction of emission could lead to a strong enhance of pollutants in the northwest? Is there some interactions between PM and dynamic lead to the re-distribution of pollutants? This may be out of the scope of this study, therefore do not expect authors' full answer here. Some discussion would be appreciated and may be an interesting topic for future study.

Reply:

- ➢ We checked the emissions from both natural sources and human activities. The reduction in emissions lead to absorption induced changes in meteorological variables, which is stronger when we use C-S model (fig. 5g). As a result, emissions in wind-blown dust have been changed in the northwest. Yes, it could be an interesting for future study.
- ➢ In the revised manuscript, we added one sentence to discuss it: "It was noted that PM2.5 concentrations was enhanced in northwest China, particularly when we used C-S model. This is related absorption-modulated natural emissions of windblown dust.

---

## Author Comment (AC2)

**#RC 2**

General comments:

The current work investigated the impacts of changes in BC mixing state during APEC on air quality and meteorology in November 2014 in Beijing. The scope of the research is of interest and well suited to the current journal, Atmos. Chem. Phys. However, the manuscript is not acceptable in its current form, because the model they used cannot answer the authors' question (please see Comment #1). The manuscript is well written and the logics is fine. It proves that the authors are certainly experts of air quality modeling and aerosol feedback processes. However, they may not well understand aerosol mixing state modeling. The simulation setup is not appropriate and so the recalculation is required. It is recommended for the authors to invite additional expert(s) for the support of relevant simulation setup. Anyway, the manuscript will be accepted in the current journal, after the authors adequately address the following comments, conduct additional simulations, and revise the manuscript accordingly. The general comments are as follows:

Reply:

➢ We thank the reviewer for careful reading and valuable suggestions, which are important to improve the quality of our manuscript.

1. The simulation setup what the authors named as "core-shell" is assuming 100% of all components mixed with BC, which never happens. This is not the case of Beijing but in Tokyo, only a small fraction (15%) of sulfate and nitrate mixed with BC (Miyakawa et al., 2014; https://doi.org/10.1080/02786826.2014.937477). Thus, there is even a possibility that the reality could be closer to what they named as "externally mixing (0% of aerosols mixed with BC)" than "core-shell (in fact 100% aerosols mixed with BC)". Anyway, the model they used cannot simulate the fractions of secondary particles mixed with BC. The reviewer recommends to use other models which can consider/resolve BC mixing state or assume fractions of BC mixing for the optical calculation of WRF-Chem that are obtained from observation in Beijing (Zhang et al., 2018) or simulation by other BC-mixing-state-considering/resolving models.

Reply:

➢ It is so good to see that the reviewer raised this issue. Yes, the "core-shell" model assumes 100% of all components mixed with BC. We knew this is not what happens in the atmosphere, as suggested by multiple observational studies (Xu et al., 2019).

➢ In summer of Beijing, the ratio of coatings to PM1 was about 25~70% (Xu et al., 2019). In winter of Beijing, the ratio of coatings to PM2.5 was about 10~40%. As you mention here, only a small fraction of sulfate and nitrate was observed to be mixed with BC in Tokyo. We stated this point in the discussion: "However, the core-shell model is an ideal scenario that assumes all non-BC materials are internally mixed and coated on BC. Zhang et al. (2016) observed that BC particles are heavily coated and are in a near-spherical shape in the North China Plain. The usage of core-shell model seems to be reasonable in this study, whereas the assumption that all non-BC materials are coated on BC might not be true in real atmosphere. The observed ratio of coatings to PM1 was ~25-70% in summer in Beijing (Xu et al., 2019), and the observed ratio of coatings to PM2.5 was ~10-40% in winter in Beijing (Wang et al., 2019)."

➢ In the revised manuscript, we added the observational study in Tokyo and added discussions on how the mixing fractions would affect the absorption of BC and its feedbacks.

➢ We conducted additional 10 sets of simulations with 10%, 20%, 40%, 60%, 80% mixing fractions of coating materials with BC. The results are shown in the added Figure 9.

➢ "We conducted additional simulations assuming 10%, 20%, 40%, 60%, and 80% mixing fractions of coating aerosols, and explored how these assumptions would affect estimated BC AAOD, and its feedbacks on radiation, boundary layer and air pollutants. As suggested in Figure 9, modelled BC AAOD increases gradually when mixing fraction rises from 0.1 to 0.4, but it keeps relatively stable when mixing fraction is between 0.4 and 1. The responses of near surface shortwave radiation and PBLH are in line with it, exhibiting relatively constant reductions when mixing fraction is higher than 0.4 (Figure 9). However, the no significant relationships are found for BC absorption induced changes in O3 (<10% difference between fractions of 0.1 and 1) and PM2.5 (<22% difference between fractions of 0.1 and 1) in Beijing. These results suggest that the findings demonstrated in this study might not be largely affected by the assumptions in mixing fractions of coating particles."

[Figure]

**Figure 9: Daytime (10:00-17:00 local time) mean BC AAOD in Beijing (a) BC absorption induced changes in shortwave radiation (b), PBLH (c), O3 (d), and PM2.5 (e) as a function of fraction of non-BC particles mixed with BC.**

➢ Although the model we used did not explicitly treated the aging process. We played with the optical calculation codes to see how it would affect our results. We hope these efforts could address the reviewers' concerns here.

Ref.:

Xu, W., Xie, C., Karnezi, E., Zhang, Q., Wang, J., Pandis, S.N., Ge, X., Zhang, J., An, J., Wang, Q. and Zhao, J., 2019. Summertime aerosol volatility measurements in Beijing, China. *Atmospheric Chemistry and Physics*, *19*(15), pp.10205-10216.

Wang, J., Liu, D., Ge, X., Wu, Y., Shen, F., Chen, M., Zhao, J., Xie, C., Wang, Q., Xu, W. and Zhang, J., 2019. Characterization of black carbon-containing fine particles in Beijing during wintertime. Atmospheric Chemistry and Physics, 19(1), pp.447-458.

Miyakawa, T., Takeda, N., Koizumi, K., Tabaru, M., Ozawa, Y., Hirayama, N., and Takegawa, N.: A new lase induced incandescence – mass spectrometric analyzer (LII-MS) for online measurement of aerosol composition classified by black carbon mixing state. Aerosol Sci. Tech., 48, 853-863, 2014.

Specific comments:

Introduction

2. 44-45: "In addition to contributing considerably to particulate matter and degraded air quality". BC usually contributes only 10% or less to the total PM mass. Can it be "considerably"?

Reply:

- We delete "considerably" here.

3. Fourth paragraph of Introduction: There are plenty of studies regarding BC mixing state in Nicole Riemer's group using PartMC/MOSAIC. For the three-dimensional modeling, the author might as well cite Matsui et al. (2012) https://doi.org/10.1029/2012JD018446, as the model domain they used are close to those of the current study. There is a mixing-state resolving model called MOSAIC-mix, Ching et al. (2016) https://doi.org/10.1002/2015JD024323, which is the same aerosol module as the authors used. I have read an article reporting that PartMC/MOSAIC is coupled with WRF-Chem. The reviewer does not know if these mixing state resolving/considering models are open to community, but if so, these can be an option for the authors to tackle with the issue raised in the study.

Reply:

- PartMC/MOSAIC has been coupled to WRF but only in a single-column model (Curtis et al., 2017).
- In paragraph four, we added these references: "In the recent decade, efforts were made also to develop models to predict the dynamic evolution of aerosol mixing states (Ching et al., 2016; Curtis et al., 2017; Matsui et al., 2013; Tian et al., 2014). For example, Matsui et al. (2013) developed a 2-D aerosol bin scheme that can resolve BC mixing state and BC aging processes. However, these approaches are far too computationally expensive for use in regional 3-D models (Barnard et al., 2010)."

Barnard, J. C., Fast, J. D., Paredes-Miranda, G., Arnott, W. P., and Laskin, A.: Technical Note: Evaluation of the WRF-Chem "Aerosol Chemical to Aerosol Optical Properties" Module using data from the MILAGRO campaign, Atmos. Chem. Phys., 10, 7325–7340, doi:10.5194/acp-10-7325-2010, 2010.

Curtis, J. H., Riemer, N., and West, M.: A single-column particle-resolved model for simulating the vertical distribution of aerosol mixing state: WRF-PartMC-MOSAIC-SCM

v1.0, Geosci. Model Dev., 10, 4057–4079, https://doi.org/10.5194/gmd-10-4057-2017, 2017.

Tian, J., Riemer, N., West, M., Pfaffenberger, L., Schlager, H., and Petzold, A.: Modeling the evolution of aerosol particles in a ship plume using PartMC-MOSAIC, Atmos. Chem. Phys., 14, 5327–5347, https://doi.org/10.5194/acp-14-5327-2014, 2014.

Ching, J., Riemer, N., and West, M.: Impacts of black carbon particles mixing state on cloud microphysical properties: sensitivity to environmental conditions, J. Geophys. Res.-Atmos., 121, 5990–6013, https://doi.org/10.1002/2016JD024851, 2016.

Matsui, H., Koike, M., Kondo, Y., Moteki, N., Fast, J. D., and Zaveri, R. A.: Development and validation of a black carbon mixing state resolved three-dimensional model: Aging processes and radiative impact, J. Geophys. Res.-Atmos., 118, 2304–2326, doi:10.1029/2012JD018446, 2013.

Method

4. Sect. 2.2: Eqs. (11)-(13) are just a general description of the calculation method of aerosol optical properties, which are less important. The last sentences of Sect. 2.2 (Ln.168-170) should be rather elaborated using equations, as it is essentially important for the current study.

Reply:

➤ Yes, eqs. (11)-(13) are less important and we added some descriptions of core-shell model:
➤ "For core-shell internal mixing, similar averaging processes are applied to the core and shell separately. The scattering efficiency, absorption efficiency and asymmetry parameter are then obtained using the core-shell Mie theory documented in Ackerman and Toon (1981). Core-shell Mie calculation requests core radius, shell radius, refractive index of core and refractive index to shell as inputs (Ackerman and Toon, 1981; Toon and Ackerman, 1981)."
➤ We elaborate with the following sentences in the revised manuscript:
➤ "The official version of WRF-Chem does not calculate optical properties of aerosols with external mixing assumption. To assess the influence of mixing with coating particles on the light absorption of BC, estimated light absorption of pure BC is required. We modified the optical calculation module in WRF-Chem so that it does not mix BC with other chemical species in the calculation of optical properties. In the calculation of optical properties with internal mixing assumption, the volume-weighted refractive index will be inputs of the Mie code. In the calculation of optical properties of BC with external mixing assumption, we only allowed BC to go through the Mie code. The mass and particle number of BC, denoted in $M_i$ and $N_i$ in each bin i (1 through 8) are computed first, and the volume ($V_i$) are obtained then by dividing by the density of BC. During the calculation of physical diameter (equation 14), other chemical species are not considered. Accordingly, BC does not mix with other chemical species, and we named it as optical properties of BC with external mixing assumption here. It should be noted that this calculation is different from treatments of external mixing in other models that all particles are separated from each other. In our calculation, BC within each bin are internally mixed, although it does not mix with other chemical species. As the purpose of this study is to explore how coating particles on BC would affect the absorption of BC, this treatment would not be an issue."

$$\blacktriangleright \quad D_i = 2(\frac{\frac{V_i}{N_i}}{\frac{4}{3}\pi})^{\frac{1}{3}} \ (14)$$

5. Judging from Eq.(14), dust is included in the simulation, which is also a light-absorber. How do the authors treat the light absorption of mineral/anthropogenic dust particles?

Reply:

- Both anthropogenic and natural dust are considered in the simulation, and the light absorption of them are treated as well. The ref. index used for dust is ref_index_oin = cmplx(1.55,0.006). We added Table S1 to describe the used index.
- Although the absorption is considered, we differentiate it from the absorption of BC (one simulation with BC absorption, and the other one without; the differences between them are the influence of BC absorption; as a result, the absorption of dust will not be a factor here).
- We calculated the BC AAOD with two sets of simulations by using the following equation (Details are listed in Table 1):

$$\Delta_{BC-Ext-NOCTL}= NOCTL_{Ext} - NOCTL_{Ext-nobc} \ (1)$$
$$\Delta_{BC-Ext-CTL}= CTL_{Ext} - CTL_{Ext-nobc} \ (2)$$
$$\Delta_{BC-CS-NOCTL}= NOCTL_{CS} - NOCTL_{CS-nobc} \ (3)$$
$$\Delta_{BC-CS-CTL}= CTL_{CS} - CTL_{CS-nobc} \ (4)$$
$$\Delta_{BC-CS-CYSN}= CYSN_{CS} - CYSN_{CS-nobc} \ (5)$$

Table S1. Complex refractive index for major aerosol components in WRF-Chem

| Components | real | imaginary |
|---|---|---|
| BC | 1.85 | 0.71 |
| Dust | 1.55 | 0.006 |
| Organics | 1.45 | 0 |
| $NH_4Cl$ | 1.50 | 0 |
| $NH_4NO_3$ | 1.50 | 0 |
| $NH_4HSO_4$ | 1.47 | 0 |
| $NA_2SO_4$ | 1.50 | 0 |

6. Sect. 2.3: Which wave lengths of AAOD (observation and simulation) did the authors use in their study?

Reply:

- We used AAOD at 440nm from observations and simulations.
- WRF-Chem model optical properties outputs are at 300nm, 400nm, 600nm and 1000nm. We derived AOD at 440nm based on Angstrom exponent relation (Schuster et al., 2006).
- We added this information in the revised manuscript: "We derived AAOD at 440nm based on Angstrom exponent relation (Schuster et al., 2006) to make it consistent with observations.".

7. Sect. 3.1: Better to compare AOD also, which might be able to partly answer why simulated AAOD was underestimated.

Reply:

  ➢ We added comparisons of AOD in the revised manuscript. Model shows between agreement with observed AOD. Although underestimation still happens, it is not as serious as AAOD.

[Figure]

**Figure 1: Scatter plots of modeled and observed near surface meteorological variables (a: T, b: RH, c: wind speed); modelled AAOD with core-shell model/external mixing assumption, and the comparisons against observations (d); modelled and observed AOD (e).**

8. Sect. 3.2: rBC is not an indicator of BC aging, but just the relative concentrations of BC and components other than BC. Aging of BC depends not only on abundance of secondary components but also on relative abundance of BC-containing and BC-free particles. If BC-free particles exist more, the secondary components condensed more toward the BC-free particles, which do not contribute aging of BC.

Reply:

  ➢ Yes, rBC is not an indicator of BC aging.
  ➢ Yet, with the 100% mixing assumption, it indicates aging. rBC values would affect the modelled enhancement of BC absorption.
  ➢ As our model does not explicitly treat aging process, we added more simulations to explore the influence of assumptions.

➢ To avoid confusion, we modified related contexts to "We used the ratio of the sum of pollutants (primary as well as secondary) to black carbon concentrations ($rBC$) to track the relative abundance of BC and non-BC particles, which is essential in the C-S calculation".

9. Sect. 3.3: E_ab is not indicting the light-absorption enhancement of BC, but indicting the difference of AAOD between 100% internal mixture with BC assumption and 0%.

Reply:

➢ Yes, it is the difference of AAOD between 100% and 0% mixture with coating particles in the core-shell model.
➢ We aim to examine how emission reduction would affect shell in core-shell model, and the resulting impacts on absorption of BC and feedbacks, not to explicitly represent the aging process. We agree that the simulation of aging process is definitely of great importance, which deserves more investigations when 3-D model becomes available.
➢ To avoid confusion, we added descriptions in the revised manuscript:
➢ "Accordingly, BC does not mix with other chemical species, and we named it as optical properties of BC with external mixing assumption here. It should be noted that this calculation is different from treatments of external mixing in other models that all particles are separated from each other. In our calculation, BC within each bin are internally mixed, although it does not mix with other chemical species."
➢ We use E_ab to represent the enhancement due to mixing with shell particles, not exactly due to internal mixing of all particles. We also added clarification in the revised manuscript: "Here we follow the method in Curci et al. (2019), and calculate $\mathrm{E}_{ab}$ as the ratio of BC AAOD estimated assuming core-shell internal mixing to that calculated with external mixing assumption (enhancement due to mixing with coating non-BC particles):".

Throughout the manuscript

10. The paper assumed clear-sky conditions for the whole analysis period. If there are no cloud during the period, please describe it. The reviewer is curious to see the simulated (with different sensitivity tests) and observed solar radiations during the period.

Reply:

➢ It is true that aerosol-radiation interactions play important roles only if clouds are not prevailing. We did see that clouds were not happening frequently during the study period in Beijing, as shown in Figure S1, and Figure S2.
➢ In the revised manuscript, we added also simulated and observed solar radiations. As shown in Figure S3, model with emission control applied shows better agreements with observations. Model with BC absorption and C-S assumption shows better agreements (r=0.95), although inconsistency still happens.

[Figure]

Figure S1. Observed cloud fraction in Beijing.

[Figure]

Figure S2. Satellite monitored clouds over East Asia.

[Figure]

Figure S3. Observed and modelled daily maximum near ground radiation.